# Female top managers and firm performance

**Inmaculada Martínez-Zarzoso** [1,2]*

**1** University of Goettingen, Goettingen, Germany, **2** University Jaume I, Castellón de la Plana, Spain

* imartin@gwdg.de

**Data Availability Statement:** Data are available from the World Bank Business Survey https://www.enterprisesurveys.org/en/enterprisesurveys.

**Funding:** Funded by Femise. I acknowledge support by the Open Access Publication Funds of the University of Goettingen and I am also grateful

## Abstract

This paper uses firm-level data worldwide to investigate productivity gaps between female and male-managed companies in developing and developed countries and compare the outcomes obtained for different regions in the world. The main aim is to shed some light on the debate around the existence of performance differences when females participate in managerial activities. The main results indicate that it is crucial to distinguish between female management and female ownership and the confluence between both. We find that when the firms have a top female manager and ownership is exclusively male, firms show higher average labor productivity. We argue that firms owned by males belong to male-dominated corporate culture and would only select a female manager if she is more competent than potential male candidates. These results are very heterogeneous among regions, of which South Saharan Africa, East Asia, and South Asia are driving the main results.

## Introduction

Globally, women's labor force participation rate is 26.5 percentage points below that of men, almost stagnating at a low rate of 48.5% [1]. When women are entrepreneurs, they tend to be "necessity" entrepreneurs, pushed by poverty and lack of other options, as opposed to being "opportunity" entrepreneurs seeking wealth accumulation [2]. Women entrepreneurs also operate in smaller firms and less profitable sectors, with less access to inputs. There is a growing consensus that gender inequality structures the relations of production in different societies [3]. Works by Amartya Sen [4–6] and a generation of feminist economists [7, 8] have pushed for an agenda focused on women's economic empowerment and the analysis of the structural barriers that women face. The Beijing Platform for Action of 1995 was one of the early global documents to promote women's economic independence [9].

Several international organizations, among them the World Bank (WB), the World Trade Organization (WTO), and the United Nations Development Program (UNDP), have introduced gender as a crucial cross-cutting issue that needs to be addressed in the fields of economics and social sciences. In particular, the World Bank has several programs targeted at boosting women's empowerment, promoting women's entrepreneurship, and improving women's health. The statement that appears at the Bank website is: "The World Bank Group takes as its starting point that no country, community, or economy can achieve its potential or meet the challenges of the 21st century without the full and equal participation of women and men, girls and boys". In addition, the achievement of gender equality and empowerment of

for the financial support of the Spanish Research Agency (PID2020-114646RB-C42 / AEI 10.13039 / 501100011033), as well as for the grant received from FEMISE (4210) at an earlier stage of this project. The funders had no role in study design, data collection and analysis, decision to publish, or preparation of the manuscript.

**Competing interests:** The authors have declared that no competing interests exist.

women is one of the commitments of the Sustainable Development Goals (SDG 5), to which the United Nations (UN) Member States committed in 2015, with a deadline of 2030 [10]. Only with males and females having equal opportunities and power could the effective use of talent by enterprises be guaranteed.

This paper uses firm-level data available worldwide from the World Bank Enterprise Survey (WBES) to investigate productivity gaps between female and male-managed companies in developing countries and compare the outcomes obtained for different regions. In developing countries in particular, it is also relevant to investigate the factors that drive gender gaps in firm performance, firm size, and access to finance. We pay special attention to differences across world regions, given that women's participation in economic activities is less prevalent in some of them. This research aims to shed some light on the debate around the existence of performance differences when females participate in managerial activities. The output of this research could help to give some insights on the appropriate programs to support women-led firms, which should be financed by international institutions, such as the WB or the UN.

Existing research on the performance gap between female and male firm owners for several world regions indicates significant gender gaps in terms of firm size but not always in terms of sales growth and productivity [11, 12]. While previous papers determine a firm's gender by whether or not there is a female owner [11–13], in this paper, we focus on the top manager being a female since the decision-maker is the manager and hence the person responsible for the performance of the firm (already pointed out by [11, 14, 15]).

Our main contribution to the literature is using newly available gender variables (2016 version of the WBES surveys) to analyze the relationship between gender and firm performance using a global sample of countries. More specifically, we investigate whether there is a gender gap in performance when the top manager is a female and compare the results with a gender gap when the ownership criterion is used to define the gender variable. The methodology used consists of regression analysis and the use of propensity score matching to construct a counterfactual.

The main results indicate that it is crucial to distinguish between female management and female ownership and the confluence between both. We find that when the firms are managed by females and ownership is exclusively male, firms show higher average labor productivity and TFP. However, if females are among the owners and a female is the top manager of a given firm, then the productivity is, in general, lower than for other firms. We argue that firms owned by males belong to a male-dominated corporate culture and would only select a female manager if she is more competent than potential male candidates. This result is in accordance with what has been documented for the United States (US) [16] and Norway [17]. These results are very heterogeneous among regions. In particular, results in Sub-Saharan Africa, East Asia, and South Asia seem to be driving the general results, whereas, in Latin America, Eastern Europe, and Central Asia, female participation in ownership seems to be negatively related to firm performance.

The rest of the paper is structured as follows; Section 2 revises the closely related literature. Section 3 describes the data, variables, and presents the stylized facts. Section 4 presents the main results and robustness checks, and finally, Section 5 concludes.

## Literature review

According to the literature on gender gaps in firm performance [11, 16], there are two main explanations for the fact that female-owned firms tend to have worse performance than male-owned firms. On the one hand, the constrained-driven gaps view indicates that females face

more constraints than males in the business environment of developing countries. For instance, it could be that access to credit is more restricted to women than to men, that legal treatment is gender-biased, or that corruption and crime affect more female entrepreneurs than male ones. In general, these gender barriers are related to gender discrimination and gender-based social norms, which is in line with the argument of the liberal feminist approach that defends the equal capacity of men and women when controlling for all potential discrimination factors.

On the other hand, the preference-driven gap explanation states that females might prefer activities in services and trade and tend to operate at a lower scale. In this case, individual choices would be responsible for lower female participation rates and female success in entrepreneurship [11]. This view overlaps with the social feminist approach, according to which males and females socialize differently, and this influences their managerial approaches. [18] name the existence of barriers to access to finance, and the business regulatory environment as potential explanations for the concentration of female entrepreneurs in low-capital intensive sectors with lower potential to grow. Similarly, [15] mainly focus on the Oaxaca decomposition analysis and find that the lower labor productivity of women-managed firms is mainly explained by women-managed firms being less protected from business environmental issues, such as crime and power outages, not having their own websites, and not being co-owned by foreigners. However, [14, 19–21] find no evidence that access to finance (or regulatory burdens) causes differences in performance between female and male-owned firms in Africa (the two first studies) and Latin America [21].

Some studies do not corroborate the hypothesis of relative female underperformance in entrepreneurship. [11], using WBES data, find that female-owned firms in Africa are at least as productive as male-owned firms, and other studies find that even female-owned firms perform better. Moreover, [12] investigates obstacles to firm growth and its links with female ownership in Latin America and Caribbean (LAC) countries. They find that female-owned firms face higher-level obstacles in relation to crime and competition, but not concerning corruption and access to finance. Moreover, they find that in terms of labor productivity, female-owned enterprises are more productive than their male counterpart and that there are not significant gender differences in terms of sales growth. The authors remark that even facing more obstacles, female-owned firms perform better or not worse than male-owned ones. These results could be connected to another literature strand that emphasizes how females have to perform better to achieve similar job status as males. For instance, [16] document that in the US retail sector, women are 14 percent less likely to be promoted relative to the sample average, despite receiving higher performance ratings on average. This evidence supports the argument made in this paper that women, in many settings, have to be disproportionately qualified to be selected for management positions. In the same line, [17] document that women are more likely to be promoted when there are more female bosses in the next higher rank, which could be connected to the second argument in this paper that women owners act as mentors for female managers.

Before moving to the empirical analysis, it is important to remark that [14] finds that the definition of female enterprise matters for the results. Most of the existent studies use a measure of female participation in ownership; however, many of these women owners have little or no involvement in the firm's management. [22] find that whereas using the 'participation in ownership' does not lead to differences in firms' performance by gender in Africa, restricting firms to those in which the women owner is the chief decision-maker does lead to a significant favorable productivity gap of 15 percent. The other authors that also show anecdotal evidence experimenting with this alternative definition are [11, 13].

## Data and variables

We use the multi-country version of the World Bank Enterprise Survey dataset (WBES) released in October 2016 [23]. The questionnaires are based on similar sampling techniques and provide fairly comparable firm-level data. It includes countries in six developing regions, namely Sub-Saharan Africa (SSA), East Asia and Pacific (EAP), South Asia Region (SAR), Eastern Europe and Central Asia (ECA), LAC, and Middle East and North Africa (MENA). In addition, the data includes two regions comprising high-income (HI) countries; one for OECD and the second for non-OECD countries. The list of countries and years of the surveys can be found in S1 Table (Supporting information), and the number of firms by region and year in S2 Table. The variables used are described in S3 Table, indicating the corresponding question and the definition of the created dummy variables.

The surveys are based on random samples constructed using stratified random sampling. Only formal (registered) companies with five or more employees are targeted for interviews, and firms with 100 percent government/state ownership are not eligible to participate in the survey. In general, business owners and top managers are interviewed, but sometimes the survey respondent calls company accountants and human resource managers into the interview to answer questions concerning the sales and labor sections of the survey. The questionnaire covers a broad range of business environment topics, including access to finance, corruption, infrastructure, crime, competition, and performance measures. Typically, 1200–1800 interviews are conducted in larger economies, 360 interviews in medium-sized economies, and only 150 interviews in small economies (See www.enterprisesurveys.org [23] for more details). For some variables, namely sales, exporting, and importing status, we are able to use the information for an additional year per questionnaire since each firm is asked in the questionnaire for the value of sales and the export/import status not only in the current but also in the previous year.

Our target variables are related to female ownership and female top managers. The question: *are any of the owners female*? (code *b4* in the dataset) allows us to identify whether there is a woman among the owners. A second question classifies firms into five categories (code *b4a_cat*) according to sex dominance in the firm's ownership. We construct a dummy that takes the value of one if ownership is equally divided among males and females, if females are a majority, or if all owners are females, zero otherwise. This variable is used as a proxy for female dominated ownership (50 percent or above). A third question asks whether the top manager is a female (code *b7a*). For this variable, there are fewer answers available, and hence the sample is restricted. The correlation between female presence in ownership and female executive is 0.42 percent. Hence, in some cases (12 percent), the top manager of firms owned by at least a female is also a female.

The data enables us to identify also a number of firm performance variables, as well as variables capturing the main obstacles that may affect the relative performance of female versus male-owned enterprises. The main performance variables we use are labor productivity and total factor productivity (TFP). Other authors use sales and employment growth as well. However, we argue that sales and number of workers are misreported and errors in the data are an important issue. Both, sales and employment growth have huge standard deviations and many missing data. Descriptive statistics corresponding to the main variables of interest are shown in S4 Table and pairwise correlations in S5 Table.

Table 1 shows that female entrepreneurs are a minority in all regions examined, but with marked differences. The table shows the average shares of females involved in entrepreneurship by world region. Three definitions of gender are considered, *fem = 1* if there is at least a female owner, *tfem = 1* if the top manager is a female, and *femmore = 1* if at least 50% of the owners are females.

**Table 1. Share of female entrepreneurs by world region.**

|  | Region | fem[a] | tfem[b] | femmore[c] |
|---|---|---|---|---|
| *mean* | **SSA**[d] | 0.29 | 0.14 | 0.16 |
| *se* |  | 0.45 | 0.34 | 0.37 |
| *Nobs* |  | 23006 | 17726 | 17360 |
| *mean* | **EAP**[f] | 0.50 | 0.27 | 0.24 |
| *se* |  | 0.50 | 0.44 | 0.43 |
| *Nobs* |  | 15755 | 14759 | 7191 |
| *mean* | **ECA**[g] | 0.36 | 0.17 | 0.17 |
| *se* |  | 0.48 | 0.38 | 0.37 |
| *Nobs* |  | 17682 | 16573 | 8459 |
| *mean* | **LAC**[h] | 0.37 | 0.16 | 0.24 |
| *se* |  | 0.48 | 0.37 | 0.43 |
| *Nobs* |  | 20576 | 12732 | 699 |
| *mean* | **MENA**[i] | 0.10 | 0.04 | 0.05 |
| *se* |  | 0.30 | 0.21 | 0.22 |
| *Nobs* |  | 6232 | 7311 | 5807 |
| *mean* | **SAR**[j] | 0.16 | 0.08 | 0.06 |
| *se* |  | 0.37 | 0.27 | 0.23 |
| *Nobs* |  | 17219 | 14596 | 12880 |
| *mean* | **OECD-HI**[k] | 0.36 | 0.17 | 0.20 |
| *se* |  | 0.48 | 0.37 | 0.40 |
| *Nobs* |  | 5996 | 5212 | 2394 |
| *mean* | **Non-OECD-HI**[l] | 0.36 | 0.21 | 0.26 |
| *se* |  | 0.48 | 0.41 | 0.44 |
| *Nobs* |  | 9314 | 8285 | 918 |
| *mean* | **Total** | 0.32 | 0.16 | 0.14 |
| *se* |  | 0.47 | 0.36 | 0.35 |
| *Nobs* |  | 115780 | 97194 | 55708 |

Note

[a]fem = 1 if at least a female is among the owners, zero otherwise

[b]tfem = 1 if the top manager is a female, zero otherwise

[c]femmore = 1 if 50% or more of the owners are females. Source: [23].

[d]SSA = Sub-Saharan Africa

[e]EAP = East Asia and Pacific

[f]ECA = Eastern Europe and Central Asia

[g]LAC = Latin America and the Caribbean

[h]MENA = Middle East and North Africa

[i]SAR = South Asia Region

[j]OECD-HI = High Income Organization for Economic Cooperation and Development

[k]Non-OECD-HI = High Income no members of the Organization for Economic Cooperation and Development.

The first column shows that the presence of females among the owners (definition of gender frequently used in previous research) is around 36 percent in ECA, a number similar to the average in high-income OECD and non-OECD countries, slightly lower in SSA (29 percent) and much lower in SAR and MENA. In contrast, EAP countries show an average share of female owners close to 50%. The second column shows the average share of female top managers (*tfem*); the shares are much lower in general and follow a similar pattern across regions and countries, with EAP countries also showing the highest average share (27%) and MENA

**Table 2. Differences in performance between male and female-owned/managed firms.** Univariate tests.

| Developing Countries | Female presence in ownership | | | Top manager is a female | | | More than 50% of female-owned firms | | |
|---|---|---|---|---|---|---|---|---|---|
| Dummy variable = | 0 | 1 | t-Stat | 0 | 1 | t-Stat | 0 | 1 | t-Stat |
| Ln sales | 16.69 | 16.95 | -11.06 | 16.98 | 16.96 | 0.68 | 16.98 | 16.70 | 6.26 |
| Ln value added per worker | 12.97 | 13.02 | -1.78 | 13.06 | 13.24 | -3.85 | 13.22 | 13.31 | -1.60 |
| Ln sales per worker | 13.41 | 13.51 | -4.77 | 13.54 | 13.79 | -8.06 | 13.67 | 13.83 | -3.88 |
| Crime | 1.18 | 1.15 | 3.24 | 1.15 | 1.13 | 1.74 | 1.03 | 1.04 | -0.91 |
| Informal | 1.48 | 1.52 | -4.80 | 1.48 | 1.46 | 1.33 | 1.39 | 1.46 | -3.98 |
| Corruption | 1.81 | 1.63 | 17.32 | 1.79 | 1.55 | 16.82 | 1.80 | 1.57 | 12.00 |
| Access to finance | 1.50 | 1.45 | 5.72 | 1.49 | 1.42 | 5.04 | 1.47 | 1.47 | -0.24 |
| Ln age | 2.57 | 2.66 | -13.84 | 2.61 | 2.54 | 8.29 | 2.57 | 2.47 | 9.44 |
| Owner concentration | 0.83 | 0.73 | 56.91 | 0.79 | 0.80 | -2.75 | 0.81 | 0.83 | -6.62 |
| Experience | 16.23 | 17.42 | -16.27 | 17.13 | 15.43 | 16.39 | 16.06 | 15.62 | 3.39 |
| Exporter | 0.20 | 0.25 | -18.13 | 0.22 | 0.21 | 2.34 | 0.20 | 0.18 | 4.51 |
| Foreign-owned | 0.08 | 0.06 | 8.23 | 0.08 | 0.06 | 6.34 | 0.07 | 0.06 | 5.40 |
| **Developed Countries** | **Female presence in ownership** | | | **Top manager is a female** | | | **More than 50% female-owned** | | |
| Dummy variable = | 0 | 1 | t-Stat | 0 | 1 | t-Stat | 0 | 1 | t-Stat |
| Ln sales | 16.50 | 16.14 | 6.75 | 16.57 | 15.65 | 13.09 | 14.49 | 13.84 | 3.43 |
| Ln value added per worker | 12.56 | 12.33 | 3.87 | 12.58 | 12.09 | 5.74 | 10.97 | 10.53 | 1.89 |
| Ln sales per worker | 13.16 | 12.86 | 7.51 | 13.13 | 12.71 | 7.81 | 11.46 | 11.39 | 0.50 |
| Crime | 1.07 | 1.21 | -4.80 | 1.10 | 1.20 | -2.74 | 0.70 | 0.77 | -0.84 |
| Informal | 1.26 | 1.37 | -3.60 | 1.22 | 1.19 | 0.90 | 1.14 | 1.04 | 0.98 |
| Corruption | 1.43 | 1.42 | 0.28 | 1.47 | 1.32 | 3.89 | 1.01 | 1.08 | -0.75 |
| Access to finance | 1.43 | 1.46 | -0.92 | 1.45 | 1.37 | 2.37 | 1.14 | 1.22 | -0.85 |
| Ln age | 2.32 | 2.47 | -7.58 | 2.40 | 2.32 | 3.36 | 2.56 | 2.48 | 1.53 |
| Owner concentration | 0.81 | 0.70 | 17.18 | 0.77 | 0.80 | -4.57 | 0.85 | 0.86 | -0.43 |
| Experience | 16.43 | 18.03 | -6.90 | 16.51 | 15.63 | 3.14 | 18.25 | 18.18 | 0.10 |
| Exporter | 0.22 | 0.21 | 0.95 | 0.21 | 0.13 | 8.06 | 0.39 | 0.17 | 6.52 |
| Foreign-owned | 0.06 | 0.04 | 3.58 | 0.05 | 0.05 | 0.80 | 0.08 | 0.02 | 3.63 |

Note: Authors' calculations using data from the WB Enterprise Surveys.

the lowest (4%). Finally, the third gender variable, female dominated ownership, is shown in column 3 for world regions. At least half of the owners are females in 24 percent of the firms in EAP, a region that shows the highest number among the developing regions, whereas the lowest share (5%) is shown for MENA countries. Since the variable female dominated is missing for half of the firms in the sample, we base the empirical analysis on the other two gender variables, namely female participation in ownership and female top manager.

The second stylized fact that has been found in previous studies is that female-owned firms tend to be smaller in size and show worse performance in terms of firm size (total revenue) and efficiency (labor productivity and value-added per worker). Table 2 shows the test results of mean differences in the performance variables and some obstacles identified by male and female-owned/managed firms for the sample of developing countries.

We find that firms with female participation in ownership are, on average higher in size (total sales) and more productive than others, whereas female dominated is associated with lower average sales, higher labor productivity, and higher value-added per employee. In addition, no significant differences in size are found for firms with female top managers, whereas their average performance is larger than for male-managed firms. However, since we expect to find heterogeneity by region, Table 3 presents similar results for each of the world regions but

**Table 3. Differences in performance between male and female-managed firms by world region.**

| Top manager: | Male | Female | t-Stat | Male | Female | t-Stat | Male | Female | t-Stat | Male | Female | t-Stat |
|---|---|---|---|---|---|---|---|---|---|---|---|---|
| **Region** | **SSA**[a] | | | **EAP**[b] | | | **ECA**[c] | | | **LAC**[d] | | |
| Ln sales | 16.77 | 16.3 | 5.86 | 19.06 | 18.53 | 7.54 | 15.96 | 15.76 | 2.72 | 16.48 | 15.53 | 10.94 |
| Ln value added per worker | 13.52 | 13.11 | 3.02 | 14.84 | 14.51 | 3.78 | 11.68 | 11.7 | -0.16 | 12.19 | 11.83 | 3.41 |
| Ln sales per worker | 13.75 | 13.68 | 0.96 | 15.26 | 15.15 | 1.72 | 12.59 | 12.78 | -2.9 | 12.77 | 12.45 | 4.31 |
| Crime | 1.27 | 1.36 | -3.04 | 0.65 | 0.72 | -3.61 | 0.94 | 1.03 | -2.96 | 1.86 | 1.95 | -2.82 |
| Informal | 1.79 | 1.85 | -1.84 | 1.1 | 1.09 | 0.33 | 1.39 | 1.43 | -1.35 | 1.92 | 1.92 | -0.01 |
| Corruption | 1.87 | 1.75 | 3.85 | 0.84 | 0.87 | -1.15 | 1.48 | 1.46 | 0.61 | 2.23 | 2.17 | 1.78 |
| Access to finance | 1.92 | 2.03 | -3.68 | 1.02 | 0.99 | 1.37 | 1.28 | 1.3 | -0.86 | 1.7 | 1.71 | -0.4 |
| Ln age | 2.45 | 2.34 | 5.89 | 2.62 | 2.59 | 1.87 | 2.46 | 2.4 | 4.39 | 2.96 | 2.84 | 5.9 |
| Owner concentration | 0.85 | 0.87 | -2.96 | 0.81 | 0.8 | 2.75 | 0.8 | 0.83 | -5.45 | 0.72 | 0.74 | -3.47 |
| Experience | 14.99 | 12.81 | 10.19 | 16.23 | 15.67 | 3.1 | 17.12 | 15.7 | 6.53 | 22.17 | 18.24 | 13.7 |
| Exporter | 0.18 | 0.16 | 1.76 | 0.24 | 0.24 | 0.18 | 0.25 | 0.19 | 6.21 | 0.29 | 0.19 | 9.79 |
| Foreign-owned | 0.13 | 0.09 | 5.77 | 0.1 | 0.06 | 6.91 | 0.06 | 0.05 | 2.27 | 0.11 | 0.07 | 5.57 |
| **Region** | **MENA**[e] | | | **Non-OECD-HI**[f] | | | **OECD-HI**[g] | | | **SAR**[h] | | |
| Ln sales | 16.14 | 15.5 | 3.47 | 16.57 | 15.65 | 13.09 | 17.58 | 16.19 | 11.08 | 17.19 | 17.89 | -10.3 |
| Ln value added per worker | 12.2 | 11.08 | 4.41 | 12.58 | 12.09 | 5.74 | 13.99 | 13.08 | 5.63 | 13.04 | 13.26 | -4.45 |
| Ln sales per worker | 12.94 | 12.34 | 3.42 | 13.13 | 12.71 | 7.81 | 14.04 | 13.09 | 9.41 | 13.65 | 13.85 | -4.53 |
| Crime | 1.56 | 1.49 | 0.8 | 1.10 | 1.20 | 2.74 | 0.96 | 1.10 | -3.23 | 0.87 | 0.81 | 1.62 |
| Informal | 1.62 | 1.51 | 1.28 | 1.22 | 1.19 | 0.90 | 1.17 | 1.26 | -1.77 | 1.11 | 1.18 | -1.89 |
| Corruption | 2.39 | 2.34 | 0.55 | 1.47 | 1.32 | 3.89 | 0.89 | 1.02 | -2.84 | 2.08 | 2.21 | -2.99 |
| Access to finance | 1.66 | 1.63 | 0.46 | 1.45 | 1.37 | 2.37 | 1.09 | 1.03 | 1.25 | 1.33 | 1.37 | -1.05 |
| Ln age | 2.64 | 2.61 | 0.71 | 2.40 | 2.32 | 3.36 | 2.90 | 2.76 | 4.89 | 2.65 | 2.65 | 0.06 |
| Owner concentration | 0.75 | 0.74 | 0.54 | 0.77 | 0.80 | -4.57 | 0.79 | 0.82 | -2.43 | 0.79 | 0.73 | 6.99 |
| Experience | 20.31 | 17.41 | 4.37 | 16.51 | 15.63 | 3.14 | 21.47 | 18.57 | 6.94 | 14.56 | 13.88 | 2.27 |
| Exporter | 0.23 | 0.22 | 0.58 | 0.21 | 0.13 | 8.06 | 0.41 | 0.27 | 8.19 | 0.15 | 0.28 | -11.63 |
| Foreign-owned | 0.05 | 0.07 | -0.97 | 0.05 | 0.05 | 0.80 | 0.12 | 0.09 | 2.18 | 0.01 | 0.02 | -3.42 |

Note: Source: [23].

[a]SSA = Sub-Saharan Africa

[b]EAP = East Asia and Pacific

[c]ECA = Eastern Europe and Central Asia

[d]LAC = Latin America and the Caribbean

[e]MENA = Middle East and North Africa

[f]OECD-HI = High Income Organization for Economic Cooperation and Development

[g]Non-OECD-HI = High Income no members of the Organization for Economic Cooperation and Development

[h]SAR = South Asia Region.

focuses specifically on the gender of the top manager. The table also includes gender differences in factors that are known to affect firm performance, such as the manager's experience, exporter status, or foreign ownership, and factors that are perceived as investment climate constraints. According to [15, 18], access to finance or crime could be a higher constraint for female than male managers. In terms of total sales, male-managed firms have higher sales than female-managed firms in most regions, except SAR. In terms of value-added per worker (labor productivity), no significant differences are found for the EAC, SSA, and EAP regions, whereas for LAC and MENA, the performance is higher for male-managed firms. In many regions, crime is perceived as a higher constraint for female managers, whereas access to finance is only perceived as a higher constraint for female managers in SSA.

Table 4. Female participation and firm size.

| | Female Top Manager | Female Presence in Ownership | Female Dominated (50% or above) | Average Number of Female Employees |
|---|---|---|---|---|
| Size Category | Developing countries | | | |
| small(<20) | 17.84% | 29.83% | 17.08% | 3 |
| medium(20–99) | 13.26% | 32.09% | 11.70% | 12 |
| large(>100) | 12.76% | 35.74% | 8.47% | 137 |
| Overall mean | 15.21% | 31.71% | 13.79% | 23 |
| | Developed countries | | | |
| small(<20) | 24.81% | 38.60% | 27.37% | 4 |
| medium(20–99) | 16.46% | 33.65% | 17.14% | 17 |
| large(>100) | 11.09% | 34.77% | 10.08% | 217 |
| Overall mean | 19.23% | 36.11% | 21.98% | 38 |

Note: Percentages (%) denote the average percent of firms in each case.

In Table 4, we show the participation of women in firms classified by employment size. It can be observed that a woman's presence in ownership is not more common in small and medium-size firms in developing countries, whereas in developed countries it is more prevalent in small firms. In contrast, the percentage of firms with female top managers is higher for small and medium-sized firms in both developing and developed countries. The average number of female employees is shown in the last column of Table 4 and indicates that the average number of women in the labor force is lower in developing than in developed countries.

The descriptive statistics in S4 Table inform the general picture concerning gender participation in entrepreneurship. However, a statistical analysis is required to investigate gender gaps while controlling for factors that could be correlated with the target variables, that is, female ownership and female manager dummy variables.

## Model specification and main results

The baseline model investigates gender gaps in performance by estimating a regression where the dependent variables are labor productivity and total factor productivity, respectively.

The first dependent variable is the sales per worker as a proxy for labor productivity. Alternatively, value-added per worker, computed as total sales minus the value of materials and intermediate inputs used in production have also been used as a proxy for labour productivity. The empirical model is given by:

$$ln\left(\frac{Sales}{nworkers}\right)_{ic}$$
$$= \alpha_0 + \alpha_{fp}fem_{ic} + \alpha_{ft}tfem_{ic} + \alpha_{int}(fem_{ic} * tfem_{ic}) + \alpha_l \ln labour_{ic} + +\sum_k \alpha_{ck} constraints_{kic}$$
$$+ \sum_j \alpha_x X_{jic} + \omega_s + \mu_c + \varphi_y$$
$$+ \eta_{ic} \tag{1}$$

where *fem* denotes female presence in ownership, it is a dummy variable that takes the value of one if among the owners there are females; *tfem* is a dummy that takes the value of one if a woman is a top manager; labor denotes the number of full-time workers, *constraints* contains several institutional factors that may constrain the performance of the firm. The variables considered are corruption, crime, competition from the informal market, and access to finance. All are measured on a scale from 1 to 4, and a higher number indicates that the corresponding variable is a very important constraint. Several controls, $X_{jic}$ have been added to the model,

including whether the firm is an exporter or is part of a multinational (partly foreign-owned), the number of years of experience of the top manager and the number of years in operation in the country. The models have also been estimated excluding a manager's experience, as suggested by an anonymous referee. It is expected that the results should be even stronger without this variable. The dependent variable, labor productivity, is measured as total sales of firm $i$ and country $c$, $sales_{ic}$, divided by the number of permanent workers, $nworkers_{ic}$. Sectoral, $\omega_s$, country, $\mu_c$, and year dummies, $\varphi_y$, are added to control for unobserved heterogeneity, and the error term, assumed to be independently and identically distributed (iid), is given by $\eta_{ic}$.

As a second main measure of productivity, we use the TFP of the firm. To calculate the TFP, we obtain estimates of a traditional Cobb-Douglas production function. The Cobb-Douglas production function is given by:

$$\ln sales_{ic} = \gamma_0 + \gamma_l \ln laborcost_{ic} + \gamma_k \ln capital_{ic} + \gamma_m \ln materials_{ic} + \omega_s + \mu_c + \varphi_y + \epsilon_{ic} \quad (2)$$

where $ln$ denotes natural logarithms, $sales_{ic}$ is total sales of firm $i$ in country c, in thousands of USD. As independent variables, we include $laborcost_{ic}$ defined as the total labor cost, including wages, salaries, and bonuses, $materials_{ic}$ denotes the total purchases of raw materials and intermediate goods, $capital_{ic}$ denotes the total fixed tangible assets of the firm, and the error term, assumed to be iid, is given by $\epsilon_{ic}$. We deflate firm-level sales and input expenditures using the industry level production price index for manufacturers for the corresponding year, and the data comes from the International Financial Statistics (IFS and UN) for manufacturing. From Eq (2), we predicted the residual and used it as dependent variable in the following model:

$$\hat{TFP}_{ic} = \beta_0 + \beta_{fp}fem_{ic} + \beta_{ft}tfem_{ic} + \beta_{int}(fem_{ic} * tfem_{ic}) + \beta_l \ln labour_{ic} + \sum_k \beta_{ck} constraints_{kic}$$

$$+ \sum_j \beta_x X_{jic} + \omega_s + \mu_c + \varphi_y$$

$$+ \varepsilon_{ic} \quad (3)$$

The interpretation of the interaction dummy is as follows. If one female is among the owners and the top manager is a male, the female owner effect is $\beta_{fp}$, and when there is a female executive, and the owners are all males, the effect of female management is $\beta_{ft}$. Finally, if $fem = 1$ and $tfem = 1$, the effects of female presence becomes $\beta_{fp} + \beta_{int}$, and the effect of female management becomes $\beta_{ft} + \beta_{int}$.

## Main results

This section shows three tables of main results. Table 5 shows the estimates obtained when labor productivity, proxied by sales per worker, is used as the dependent variable. As an alternative, value-added per worker is also used as a proxy for labour productivity. However, the sample is considerably reduced due to missing observations in the variable total purchases of raw materials and intermediate goods, which is needed to calculate value added. The results, which remain similar, are shown in S6 Table. In Table 6 TFP is used, instead of sales per worker, and Table 7 shows results using labor productivity as in Table 5, but by world region.

Table 5 shows that female presence in ownership is associated with 7.2% lower labor productivity (column 1). When adding the female executive dummy in column (2), female presence in ownership is associated with 5.8% lower labor productivity, whereas firms with female top managers show a 4.4% lower productivity than the rest. However, column (3) indicates that firms with female top managers in which there are no female owners are on average 19.1 percent more productive than male managed firms, whereas if females are among the owners and the top manager is a female, the average labor productivity is around 16 percent lower

**Table 5. Gender bias and firm performance.** Results for labour productivity.

| Dep. Variable: | (1) | (2) | (3) | (4) |
|---|---|---|---|---|
| | | Labour Productivity | | |
| **Ind. Variables** | | | | |
| Female Presence | -0.072*** | -0.058*** | -0.000 | -0.001 |
| | (0.014) | (0.015) | (0.016) | (0.016) |
| Female Top Manager | | -0.044** | 0.191*** | 0.194*** |
| | | (0.019) | (0.034) | (0.034) |
| Female Presence*Top Manager | | | -0.349*** | -0.351*** |
| | | | (0.040) | (0.040) |
| Experience of the manager | | | | 0.001* |
| | | | | (0.001) |
| Ln number of workers | 0.048*** | 0.048*** | 0.044*** | 0.044*** |
| | (0.008) | (0.008) | (0.008) | (0.008) |
| Crime | -0.007 | -0.007 | -0.007 | -0.007 |
| | (0.006) | (0.006) | (0.006) | (0.006) |
| Informal competition | -0.023*** | -0.023*** | -0.024*** | -0.024*** |
| | (0.005) | (0.005) | (0.005) | (0.005) |
| Corruption | 0.024*** | 0.024*** | 0.023*** | 0.023*** |
| | (0.005) | (0.005) | (0.005) | (0.005) |
| Access to finance | -0.057*** | -0.057*** | -0.057*** | -0.057*** |
| | (0.006) | (0.006) | (0.006) | (0.006) |
| Ln age | 0.068*** | 0.068*** | 0.067*** | 0.061*** |
| | (0.009) | (0.009) | (0.009) | (0.010) |
| Ownership concentration | -0.379*** | -0.375*** | -0.363*** | -0.361*** |
| | (0.026) | (0.026) | (0.026) | (0.026) |
| Exporter | 0.249*** | 0.249*** | 0.247*** | 0.247*** |
| | (0.019) | (0.019) | (0.019) | (0.019) |
| Foreign owned | 0.491*** | 0.491*** | 0.486*** | 0.488*** |
| | (0.032) | (0.032) | (0.032) | (0.032) |
| Observations | 60,937 | 60,937 | 60,937 | 60,937 |
| Adjusted R-squared | 0.763 | 0.763 | 0.763 | 0.763 |

Note: Robust standard errors in parentheses cluster by survey weights.

*** p<0.01

** p<0.05

* p<0.1.

Country, sector, and year dummies are added in all models, not reported to save space. Labour productivity is defined as sales per worker.

(0.191–0.349 = -0.158) according to column (3). Finally, in column (4), we added the experience of the top manager as a control variable, and the results remain practically unchanged. We have expected that the results concerning *tfem* were possibly weaker when adding experience of the manager as a control variable, but this is not the case.

A different specification (Eq 3) with TFP as the dependent variable is estimated in Table 6. The results are similar in terms of sign and significance as in Table 5 but smaller in magnitude. In the latter case, firms with a female top manager show a 9.8 percent higher labor productivity than those without, when no females are among the owners (column 3), whereas if females are among the owners and the top manager is a female, the average labor productivity is only around 1.2 percent lower (0.098–0.110 = -0.012) according to column (3). The combined effect is statistically significant at the 5 percent level. A different regression has been estimated to directly obtain the

**Table 6. Gender bias and firm performance.** Results for TFP.

| | (1) | (2) | (3) | (4) |
|---|---|---|---|---|
| Dep. Variable: | **Total Factor Productivity** | | | |
| Ind. Variables | | | | |
| Female Presence | -0.000 | -0.006 | 0.009 | 0.010 |
| | (0.016) | (0.016) | (0.019) | (0.019) |
| Female Top Manager | | 0.024 | 0.098** | 0.095** |
| | | (0.020) | (0.039) | (0.039) |
| Female Presence*Top Manager | | | -0.110** | -0.109** |
| | | | (0.044) | (0.044) |
| Experience of the manager | | | | -0.001* |
| | | | | (0.001) |
| Ln number of workers | 0.062*** | 0.062*** | 0.061*** | 0.060*** |
| | (0.006) | (0.006) | (0.006) | (0.006) |
| Crime | 0.007 | 0.007 | 0.007 | 0.007 |
| | (0.006) | (0.006) | (0.006) | (0.006) |
| Informal competition | -0.011** | -0.011** | -0.012** | -0.011** |
| | (0.005) | (0.005) | (0.005) | (0.005) |
| Corruption | 0.004 | 0.003 | 0.003 | 0.003 |
| | (0.005) | (0.005) | (0.005) | (0.005) |
| Access to finance | -0.035*** | -0.035*** | -0.035*** | -0.036*** |
| | (0.006) | (0.006) | (0.006) | (0.006) |
| Ln age | 0.004 | 0.004 | 0.004 | 0.009 |
| | (0.008) | (0.008) | (0.008) | (0.009) |
| Ownership concentration | -0.087*** | -0.089*** | -0.086*** | -0.088*** |
| | (0.025) | (0.025) | (0.025) | (0.025) |
| Exporter | 0.079*** | 0.079*** | 0.078*** | 0.079*** |
| | (0.016) | (0.016) | (0.016) | (0.016) |
| Foreign owned | 0.079*** | 0.079*** | 0.077*** | 0.075*** |
| | (0.028) | (0.028) | (0.028) | (0.028) |
| Observations | 21,887 | 21,887 | 21,887 | 21,887 |
| Adjusted R-squared | 0.031 | 0.031 | 0.031 | 0.032 |

Note: Robust standard errors in parentheses cluster by survey weights.

*** p<0.01

** p<0.05

* p<0.1. Country, sector, and year dummies are added in all models, not reported to save space. TFP = total factor productivity estimated as a residual from the production function given by Eq (2).

coefficient of the combined effect and the corresponding standard error. We argue that firms owned by males belong to a male-dominated corporate culture and would only select a female manager if she is more competent than potential male candidates. However, in firms with a more female presence in ownership, female owners could act as mentors for women managers, favouring them against potential—perhaps also more qualified—male candidates. The results in column (4) indicate that when we control for experience of the manager the estimated coefficient for *tfem* slightly decreases in magnitude, according to expectations. Arguing that the manager's experience is one proxy measure of qualification, it could be expected that the results are stronger if the regressions did not control for the manager's experience, which is the case.

Concerning the business constraints, informal competition and access to finance are statistically significant and indicate that when firms perceive the given obstacle as a

**Table 7. Gender bias in labour productivity by world region.**

| Dep. Variable: Labor Productivity | (1) | (2) | (3) | (4) | (5) | (6) | (7) | (8) |
|---|---|---|---|---|---|---|---|---|
| Country Groups: | SSA[a] | EAP[b] | ECA[c] | LAC[d] | MENA[e] | SAR[f] | OECD-HI[g] | Non-OECD-HI[h] |
| Ind. Variables | | | | | | | | |
| Female Presence | 0.099* | -0.092* | -0.082** | 0.020 | 0.226*** | 0.088** | -0.074 | -0.040 |
| | (0.053) | (0.050) | (0.035) | (0.027) | (0.077) | (0.043) | (0.049) | (0.042) |
| Female Top Manager | 0.252** | 0.345*** | -0.023 | 0.092 | -0.048 | 0.364*** | -0.016 | 0.029 |
| | (0.105) | (0.097) | (0.081) | (0.068) | (0.177) | (0.067) | (0.122) | (0.084) |
| Female Presence*Top Manager | -0.524*** | -0.385*** | -0.125 | -0.341*** | 0.027 | -0.485*** | -0.183 | -0.179* |
| | (0.126) | (0.114) | (0.091) | (0.078) | (0.277) | (0.094) | (0.142) | (0.099) |
| Ln number of workers | 0.014 | 0.028 | 0.008 | 0.126*** | 0.001 | 0.029 | 0.005 | 0.032** |
| | (0.024) | (0.029) | (0.013) | (0.012) | (0.025) | (0.019) | (0.021) | (0.016) |
| Crime | -0.052*** | 0.013 | -0.003 | 0.015 | 0.014 | -0.013 | -0.011 | -0.001 |
| | (0.019) | (0.021) | (0.012) | (0.010) | (0.019) | (0.026) | (0.019) | (0.014) |
| Informal competition | -0.053*** | 0.006 | -0.006 | -0.051*** | 0.029* | -0.013 | -0.046*** | -0.050*** |
| | (0.017) | (0.016) | (0.010) | (0.009) | (0.017) | (0.014) | (0.016) | (0.013) |
| Corruption | 0.014 | 0.038** | 0.022** | 0.012 | -0.013 | 0.023* | 0.017 | 0.030** |
| | (0.017) | (0.016) | (0.011) | (0.010) | (0.018) | (0.013) | (0.018) | (0.012) |
| Access to finance | -0.039** | -0.104*** | -0.018* | -0.065*** | -0.108*** | -0.065*** | -0.056*** | -0.007 |
| | (0.019) | (0.017) | (0.010) | (0.011) | (0.020) | (0.017) | (0.018) | (0.013) |
| Ln age | 0.184*** | 0.187*** | -0.029 | 0.077*** | 0.001 | 0.014 | 0.021 | 0.031 |
| | (0.036) | (0.031) | (0.022) | (0.019) | (0.030) | (0.022) | (0.032) | (0.027) |
| Ownership concentration | -0.492*** | -0.518*** | -0.132** | -0.110** | -0.435*** | -0.584*** | -0.203** | -0.174*** |
| | (0.114) | (0.083) | (0.055) | (0.045) | (0.088) | (0.069) | (0.090) | (0.063) |
| Experience of the manager | 0.006* | -0.001 | -0.001 | -0.001 | 0.003 | 0.003* | -0.004* | 0.000 |
| | (0.003) | (0.002) | (0.001) | (0.001) | (0.002) | (0.002) | (0.002) | (0.002) |
| Exporter | 0.026 | 0.306*** | 0.274*** | 0.258*** | 0.231*** | 0.314*** | 0.233*** | 0.297*** |
| | (0.062) | (0.067) | (0.040) | (0.034) | (0.067) | (0.053) | (0.051) | (0.044) |
| Foreign owned | 0.721*** | 0.306*** | 0.421*** | 0.462*** | 0.175 | 0.274 | 0.481*** | 0.636*** |
| | (0.084) | (0.086) | (0.080) | (0.059) | (0.112) | (0.197) | (0.096) | (0.087) |
| Observations | 8,580 | 8,574 | 10,765 | 8,506 | 4,154 | 12,225 | 3,081 | 5,052 |
| Adjusted R-squared | 0.643 | 0.799 | 0.773 | 0.850 | 0.805 | 0.136 | 0.818 | 0.610 |

Note: Robust standard errors in parentheses cluster by survey weights.

*** p<0.01

** p<0.05

* p<0.1.

Country, sector, and year dummies are added in all models, not reported to save space.

[a]SSA = Sub-Saharan Africa

[b]EAP = East Asia and Pacific

[c]ECA = Eastern Europe and Central Asia

[d]LAC = Latin America and the Caribbean

[e]MENA = Middle East and North Africa

[f]SAR = South Asia Region

[g]OECD-HI = High Income Organization for Economic Cooperation and Development

[h]Non-OECD-HI = High Income no members of the Organization for Economic Cooperation and Development. Labor productivity is defined as sales per worker.

higher constraint, this is associated with a lower performance. Firms perform better when they are exporters and have foreign owners, as has also been confirmed in the related literature.

Next, in Table 7, we show similar estimations for each region using labor productivity as the dependent variable (Eq 1). The number of observations is considerably restricted for materials and inputs and also for capital. Therefore, for the regional and country analysis, we focus on labor productivity (total sales per employee) as the dependent variable, as in Table 5. The gender variables show very heterogeneous estimated coefficients, indicating the particularities of each geographical location of the corresponding countries. The first interesting outcome is that female presence in ownership when the top manager is a male shows a positive and significant effect on labor productivity in SSA, MENA, and SAR regions and a negative effect in ECA and in EAP, being the second only marginally significant. Secondly, when a female is a top manager and the owners are all males, firms seem to show a higher performance in SSA, EAP, and SAR. However, if the top manager is a female and there is at least one female among the owners, this is associated with a lower performance in SSA, EAP and LAC, and SAR. Finally, in high income countries (columns 7 and 8) none of the relevant dummy variables is statistically significant, indicating that there is no evidence of labor productivity being different for firms managed by males and females or when the gender composition of the management team differs.

## Robustness checks

A very efficient and commonly used method to control for endogeneity problems in non-experimental and experimental causal studies is propensity score matching (PSM) in combination with regression analysis. This technique estimates the likelihood of receiving a treatment for all observations and matches each treated observation (top female manager in this paper) with one or several untreated observations (the control group: male-managed firms) according to their propensity scores. The propensity score should include only the variables that influence both the participation decision and the outcome variable (firm performance). The following logit model is estimated:

$$\text{logit}(\text{TFEM}_{ic}) = \phi(\beta_0 + \beta_l \ln labour_{ic} + \beta_k \ln capital_{ic} + \beta_m \ln materials_{ic} + \sum_k \beta_{ck} \, constrains_{kic} + \sum_j \beta_x X_{jic} + \eta_{ic}) \quad (4)$$

The PSM results generate a matched sample containing the firms with similar characteristics, and model (1) is re-estimated for the matched sample. S7 Table shows the comparison in means before and after matching and the corresponding reduction in bias. Table 8 shows that, on average, firms with top female managers have labor productivity (sales or value-added per employee), which is around 23 percent (28 percent) higher than firms with top male managers when there are no females among the owners. When we take TFP, the corresponding estimated effect is 0.14, and hence the positive difference in performance is half that it was before, but the sample has fewer observations than the one for value-added or sales per worker, and hence the results have to be interpreted with caution.

Several additional robustness were considered. The first consisted in adding the average years of education of female workers and the number of female employees as control variables, and the results hold. Results are shown in S8 Table, columns 1 and 2. As second robustness, we allowed for heterogeneous coefficients by size (See columns 3 to 5 of S8 Table). The results hold for large and medium enterprises but not for small firms. Finally, the model was estimated for the matched firms for manufacturing and services separately, and it seems to be the former sector driving the results.

## Conclusions

This paper investigates whether female participation in entrepreneurship, as owners or as main managers, is related to firm performance. Gender differences in a firm's performance

**Table 8. Results for the matched sample.**

|  | (1) | (2) | (3) |
|---|---|---|---|
| Dep. Variables: | Ln (Sales/worker) | ln (VA/worker) | TFP |
| Ind. Variables |  |  |  |
| Female Presence | -0.017 | -0.006 | 0.002 |
|  | (0.097) | (0.132) | (0.059) |
| Female Top Manager | 0.231*** | 0.278** | 0.141*** |
|  | (0.086) | (0.114) | (0.053) |
| Female Presence*Top Manager | -0.269** | -0.317* | -0.115 |
|  | (0.123) | (0.174) | (0.078) |
| Observations | 18,663 | 9,110 | 5,922 |
| Adjusted R-squared | 0.086 | 0.123 | 0.901 |

Note: Clustered Robust standard errors in parentheses.

*** $p < 0.01$

** $p < 0.05$

* $p < 0.1$.

have been investigated for different regions in the world economy using a number of proxies to measure gender participation. We depart from the existent literature by using a more comprehensive dataset available for countries in different world regions, including developed and developing countries. The second departure is using the variable top female manager as the main proxy to measure female participation in ownership and compare the results with those obtained for the most commonly used proxy, which has been a female presence in ownership.

The main results indicate that it is crucial to distinguish between female management and female ownership and the confluence between both. We find that when the firms are managed by females and there are no female owners, they show higher average labor productivity and TFP. However, if females are among the owners and a female is the top manager, their productivity is lower than in other firms. We argue that firms owned by males belong to a men-dominated corporate culture and would only select a female manager if she is more competent than potential male candidates. However, in firms with more female presence in ownership, female owners could act as mentors for women managers, favouring them against potential male candidates. In those cases, the female managers could have reached the position. These results are very heterogeneous among regions in the world economy. In particular, results in Sub-Saharan Africa, East Asia, and South Asia seem to be driving the main results, whereas, in Latin America, Eastern Europe, Central Asia, and MENA, we do not find significant differences in labor productivity between firms with top female and top male managers.

We conclude that it should be extremely desirable to dedicate more resources to educating younger generations so that gender inequality does not persist and that gender discrimination is turned around. It has been shown in this paper that female management is not necessarily associated with worse firm performance; on the contrary, it is, in specific cases, associated with an average higher labor productivity.

More research is needed for specific countries using richer datasets to relate our results to the particular business environments and cultural and social norms that are present in each country. Further research would also be desirable to investigate whether firms managed by females face higher business environment obstacles.

## Supporting information

**S1 Table. List of countries and years surveyed by world region.**
(DOCX)

**S2 Table. Number of firms surveyed by year and region.**
(DOCX)

**S3 Table. Variables definitions.**
(DOCX)

**S4 Table. Summary statistics.**
(DOCX)

**S5 Table. Pairwise correlations.**
(DOCX)

**S6 Table. Gender bias and firm performance.** Results for value added per worker.
(DOCX)

**S7 Table. Reduction bias before and after matching.**
(DOCX)

**S8 Table. Additional robustness checks.**
(DOCX)

## Acknowledgments

I would like to thank the editor and reviewers for their very helpful suggestions, as well as the participants in the seminar given at WIDER in Helsinki, the EFAS conference in Berlin and the GLAD conference in Goettingen for their helpful comments.

## Author Contributions

**Conceptualization:** Inmaculada Martínez-Zarzoso.

**Data curation:** Inmaculada Martínez-Zarzoso.

**Formal analysis:** Inmaculada Martínez-Zarzoso.

**Funding acquisition:** Inmaculada Martínez-Zarzoso.

**Investigation:** Inmaculada Martínez-Zarzoso.

**Methodology:** Inmaculada Martínez-Zarzoso.

**Project administration:** Inmaculada Martínez-Zarzoso.

**Resources:** Inmaculada Martínez-Zarzoso.

**Software:** Inmaculada Martínez-Zarzoso.

**Validation:** Inmaculada Martínez-Zarzoso.

**Writing – original draft:** Inmaculada Martínez-Zarzoso.

**Writing – review & editing:** Inmaculada Martínez-Zarzoso.

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
