## [Decision Letter · Decision Letter 0]

11 Apr 2022

PONE-D-22-04064FEMALE TOP MANAGERS AND FIRM PERFORMANCE IN DEVELOPING COUNTRIESPLOS ONE

Dear Dr. Martínez-Zarzoso,

Thank you for submitting your manuscript to PLoS ONE. I have experienced immense difficulty trying to find referees for your paper. Now, I have heard from a referee and based on the referee report and my own reading of the paper I would like to invite you to submit a revised version of the manuscript that addresses the points raised by the referee. If you choose to revise your paper, please, make sure that you pay attention to publication criteria of PLoS ONE. 

We look forward to receiving your revised manuscript.

Kind regards,

Joanna Tyrowicz

Academic Editor

PLOS ONE

Journal Requirements:

Reviewers' comments:

Reviewer's Responses to Questions

**Comments to the Author**

1. Is the manuscript technically sound, and do the data support the conclusions?

Reviewer #1: Partly

2. Has the statistical analysis been performed appropriately and rigorously? 

Reviewer #1: Yes

3. Have the authors made all data underlying the findings in their manuscript fully available?

Reviewer #1: Yes

4. Is the manuscript presented in an intelligible fashion and written in standard English?

Reviewer #1: No

5. Review Comments to the Author

Reviewer #1: The submitted paper uses firm-level data from the World Bank’s Enterprise Survey program to analyze differences in various measures of productivity between firms managed by men and women. An innovation of the paper is that it compares female ownership to female management, and investigates the interaction between the two. The paper concludes that firms with a female manager have higher levels of productivity if there is at least one woman among the firm’s owners. In cases where the firm is solely owned by women, female management is associated with lower levels of firm productivity. The author argues that this reflects that firms in male dominated corporate cultures would only select a female manager if she is more competent/qualified than potential male candidates, while firms with female representation in ownership may be more likely to support female managers.

This paper sheds light on an important, yet understudied, aspect of gender differences in firm performance – the distinction between female ownership and management and the interaction between the two. The main results of the paper are interesting and relevant.

Nonetheless, I have several concerns about the paper that ought to be addressed before considering publication. My main concern is that the paper, despite presenting interesting results, does not tell a coherent story. For example, the regional focus on the Middle East and North Africa (MENA) at the end of the paper is rather ad hoc and not tied to the paper’s main storyline, since the main empirical pattern of interest (i.e., the positive significant interaction between female top management and female ownership) is not significant for the MENA region. Likewise, the paper makes no attempt to tie its main hypothesis (i.e., that in male-dominated corporate cultures, female candidates need to be more qualified than male candidates to be selected to top management positions) to other related literature strands. Second, the paper is somewhat sloppy in terms of variable definitions, use of acronyms, consistency between the text and Tables, typos, etc. and this ought to be addressed prior to publication.

Main comments:

1. Abstract: The author states that “when the firms have a top manager and ownership is male dominated, firms show higher average labor productivity”. This statement does not seem to be supported by the author’s analysis, since the variable relating to firm ownership included in the regression model is “whether there is at least one female among the owners” (fem). This dummy variable hence does not indicate whether ownership is male-dominated, only if there is at least one woman among the owners. The above sentence should be rephrased to “and ownership is exclusively male”.

2. Introduction:

a. The introduction is too long and includes long digressions that are only tangentially relevant to the paper (e.g., a lengthy description of the World Bank’s WEFI initiative). The introduction should be streamlined and focus more (and earlier) on the main contribution of the paper (i.e., the interaction between female ownership and management).

b. Both the introduction and the title suggest that the analysis is based on data from developing countries only. This, however, is not true since the sample also includes high income countries. This should be clarified in the text.

3. Section 3 (data and variables):

a. The paper constructs a dummy variable that takes the value of one if at least half of the owners are female. This dummy variable is subsequently labeled as gender diversity. In my view, this label is misleading – since a firm where all owners are women is not necessarily “gender diverse”. A label such as “female dominated (50% or above)” would be more accurate.

b. The paper seems to distinguish between labor productivity (which is defined, as per the description in section 4, as sales per worker) and value added per worker (e.g., in the following statement: “The main performance variables we use are labor productivity and value added per employee”). However, value added per worker is also commonly used as a proxy for labor productivity (e.g., World Bank – metadata glossary: “value added per worker is a measure of labor productivity—value added per unit of input.”). Moreover, some of the studies cited by the author (e.g., Aterido and Hallward-Driemeier 2011) use value added per worker as the key measure of labor productivity. It would be more coherent to consider both sales per worker and value added per worker as measures of labor productivity and to distinguish between labor and total factor productivity.

4. Section 4 (model specification and main results)

a. The selection of control variables should be motivated better. For example, I don’t quite understand why the regression controls for the experience of the manager. The authors argue that their results reflect that firms with male owners only select female candidates if the latter are more qualified than comparable male candidates. Arguably, the manager’s experience is one (proxy) measure of qualification. I would therefore expect that, if the author’s hypothesis is correct, the results would be even stronger if the regressions did not control for the manager’s experience (and in fact, this would be a useful addition to the paper – see also point (d) below).

b. The numeric example quoted to illustrate the interaction effect does not match the Table. E.g., the text says “the average labor productivity is around 15 percent lower (0.219 – 0.37 = 0.152). However, as per Table 5, this should be “0.223 – 0.381 = 0.158 (=16%).

c. The regional focus on MENA at the end of section 4.1 appears ad hoc and not well motivated. I understand that MENA has the lowest share of female entrepreneurs by region (as per Table 1), but the main mechanism of interest documented in the paper (i.e., the positive significant interaction between female top management and female ownership) is not significant for MENA countries. Moreover, at the country level, there are no firms with both a female top manager and female presence in ownership in Jordan and Yemen, which makes it impossible to estimate the interaction effect. I would be concerned that in some of the other countries, too, the interaction effect may be based on a small number of observations. For these reasons, the MENA deep dive provides little value added to the paper’s main storyline. Instead of zooming in on one region, it would be more useful to conduct additional analysis that could provide empirical support to the notion that women managers in firms with exclusive male ownership are disproportionately competent/qualified but not necessarily in firms with a female co-owner. One option would be to compare years of experience of male and female managers in firms with exclusive male ownership compared to firms with a female co-owner, overall and by region, and explore whether these descriptive patterns are consistent with the paper’s main hypothesis.

d. I think the paper would be considerably strengthened by attempting to connect the main results to other literature strands that are related to the key hypothesis of the paper. For example, Benson et al (2021) document that in the US retail sector women are 14 percent less likely to be promoted relative to the sample average, despite receiving higher performance ratings on average (e.g., to support the author’s argument that women, in many settings, have to be disproportionately qualified to be selected into management positions). Kunze and Miller (2017) document that women are more likely to be promoted when there are more female bosses in the next higher rank (which could be connected to the author’s argument that women owners act as mentors for female managers). I am not advocating for these particular studies to be discussed in the paper, but I do think the paper would be strengthened by providing some evidence from other contexts for the hypothesized mechanisms.

Minor comments:

5. Introduction: “Existing research on the performance gap between female and male firm’ owners for other regions” -> what is meant by “other regions”?

6. Section 3: “It could be argued that for female managers access to finance or crime could be a higher constraint than for male managers” -> I believe this is a hypothesis, not a result/description of the Table (because as pointed out subsequently, only Sub-Saharan Africa shows a significant gender gap in access to finance). This distinction should be made clearer. Also, rather than saying “it could be argued” it would be preferable to cite other studies that have made this argument.

7. Section 4: The text says that sales are expressed in Egyptian pounds. Is this correct? Because this would be a rather unusual reference currency for a global dataset.

8. Regions and acronyms: The term “South Saharan Africa” is not commonly used and should be revised to “Sub-Saharan Africa”. Likewise, the paper should use consistent acronyms and labels for world regions. For example, Latin America and the Caribbean is sometimes abbreviated LA (in section 2) and some LAC (subsequently). Sub-Saharan Africa is sometimes abbreviated SSA (mostly in the text) and sometimes AFR (in the tables). Acronyms should be spelled out on first use (which is currently not the case).

9. The paper contains a lot of typos and would benefit from rigorous copy-editing. Some examples below (mainly from the paper’s first pages – but this continues throughout the paper)

a. “Constraint” and “constrain” are not equivalent (the first is a noun, the second one a verb). Throughout the paper, the two terms are conflated (e.g., section 2: “females face more constrains than males” -> this should be “constraints”; section 4: “factors that may constraint the performance” -> this should be “constrain”).

b. Introduction:

• “and that should be finance by the WEFI” -> financed (not finance)

• “our main contribution to the literature is the use newly available” -> of newly available

• “the methodology used consist on” -> consists

• “section 2 revise the closely related literature” -> reviews

c. Section 2

• “affect more females entrepreneurs that male entrepreneurs” –> more female entrepreneurs than male entrepreneurs

• “females socialize differently and this influence their” -> influences

References

Benson, Alan, Danielle Li, and Kelly Shue. 2021. “Potential and the Gender Promotion Gap”; https://danielle-li.github.io/assets/docs/PotentialAndTheGenderPromotionGap.pdf

Kunze, Astrid and Amalia R. Miller. 2017. “Women Helping Women? Evidence from Private Sector Data on Workplace Hierarchies.” The Review of Economics and Statistics 99(5): 769-75.

World Bank. Metadata glossary. https://databank.worldbank.org/metadataglossary/world-development-indicators/series/NV.AGR.EMPL.KD

6. PLOS authors have the option to publish the peer review history of their article (what does this mean?). If published, this will include your full peer review and any attached files.

Reviewer #1: No

---

## [Author Response · Author response to Decision Letter 0]

7 Jun 2022

Dear Editor and reviewer,

Thank you for inviting me to revise and resubmit the manuscript to PLoS ONE. In what follows you will find detailed answers to each referee comment as well as indications of how the suggestions have been adopted in the revised version of the paper, in which I also paid attention to publication criteria of PLoS ONE. Moreover, the new manuscript was revised by an English proof-reader to amend any typos or grammar errors.

Sincerely,

Inmaculada Martínez-Zarzoso

Review Comments to the Author

Reviewer #1: The submitted paper uses firm-level data from the World Bank’s Enterprise Survey program to analyze differences in various measures of productivity between firms managed by men and women. An innovation of the paper is that it compares female ownership to female management, and investigates the interaction between the two. The paper concludes that firms with a female manager have higher levels of productivity if there is at least one woman among the firm’s owners. In cases where the firm is solely owned by women, female management is associated with lower levels of firm productivity. The author argues that this reflects that firms in male dominated corporate cultures would only select a female manager if she is more competent/qualified than potential male candidates, while firms with female representation in ownership may be more likely to support female managers. This paper sheds light on an important, yet understudied, aspect of gender differences in firm performance – the distinction between female ownership and management and the interaction between the two. The main results of the paper are interesting and relevant.

Nonetheless, I have several concerns about the paper that ought to be addressed before considering publication. My main concern is that the paper, despite presenting interesting results, does not tell a coherent story. For example, the regional focus on the Middle East and North Africa (MENA) at the end of the paper is rather ad hoc and not tied to the paper’s main storyline, since the main empirical pattern of interest (i.e., the positive significant interaction between female top management and female ownership) is not significant for the MENA region. Likewise, the paper makes no attempt to tie its main hypothesis (i.e., that in male-dominated corporate cultures, female candidates need to be more qualified than male candidates to be selected to top management positions) to other related literature strands. Second, the paper is somewhat sloppy in terms of variable definitions, use of acronyms, consistency between the text and Tables, typos, etc. and this ought to be addressed prior to publication.

General answer:

Dear reviewer, many thanks for taking time reading the paper and for your very helpful comments and suggestions, most of which I have taken on board to revise the draft, as indicated below. First, I agree with the concern related to the focus on the MENA region in the last part of the paper. Therefore, this part has been dropped. Second, an attempt has been made to tie the main hypothesis to other related literature strands, according to your specific comment below. Finally, thought the text, in the revised draft, variable definitions have been unified, as well as acronyms and proof-reading by an English native-speaker have eliminated typos or gramma mistakes.

Main comments:

1. Abstract: The author states that “when the firms have a top manager and ownership is male dominated, firms show higher average labor productivity”. This statement does not seem to be supported by the author’s analysis, since the variable relating to firm ownership included in the regression model is “whether there is at least one female among the owners” (fem). This dummy variable hence does not indicate whether ownership is male-dominated, only if there is at least one woman among the owners. The above sentence should be rephrased to “and ownership is exclusively male”.

Answer: Many thanks for your suggestion, we have rephrased the sentence accordingly and replaced by “and ownership is exclusively male” as suggested.

2. Introduction:

a. The introduction is too long and includes long digressions that are only tangentially relevant to the paper (e.g., a lengthy description of the World Bank’s WEFI initiative). The introduction should be streamlined and focus more (and earlier) on the main contribution of the paper (i.e., the interaction between female ownership and management).

Answer: Some less-relevant paragraphs of the introduction have been dropped following your suggestion, thanks. The longer ones deleted are:

“Among the initiatives to support women in entrepreneurial activities, the WB has launched an initiative (Women Entrepreneurs Finance Initiative, WEFI) that will enable more than 1 billion USD in financing to provide technical assistance, access to credit and to invest in programs supporting women-led small and medium firms. The initiative was proposed in early 2017 by the United States and Germany and received strong support from other Development Assistance Committee (DAC) donors.”

And:

 “Given that talent is generally scarce in developing countries, discriminatory practices should be avoided because those will impede the best use of talent to the detriment of economic development.”

b. Both the introduction and the title suggest that the analysis is based on data from developing countries only. This, however, is not true since the sample also includes high income countries. This should be clarified in the text.

Answer: the title has been modified, dropping “developing countries”. Moreover, we have shortened the introduction and focused earlier on the main contribution. Likewise, we refer to the sample of countries indicating that also high-income countries are considered.

3. Section 3 (data and variables):

a. The paper constructs a dummy variable that takes the value of one if at least half of the owners are female. This dummy variable is subsequently labeled as gender diversity. In my view, this label is misleading – since a firm where all owners are women is not necessarily “gender diverse”. A label such as “female dominated (50% or above)” would be more accurate.

b. The paper seems to distinguish between labor productivity (which is defined, as per the description in section 4, as sales per worker) and value added per worker (e.g., in the following statement: “The main performance variables we use are labor productivity and value added per employee”). However, value added per worker is also commonly used as a proxy for labor productivity (e.g., World Bank – metadata glossary: “value added per worker is a measure of labor productivity—value added per unit of input.”). Moreover, some of the studies cited by the author (e.g., Aterido and Hallward-Driemeier 2011) use value added per worker as the key measure of labor productivity. It would be more coherent to consider both sales per worker and value added per worker as measures of labor productivity and to distinguish between labor and total factor productivity.

Answer: The two points a. and b. have been taken into account, thanks. First, the variable has been renamed and second, the proposed distinction has been adopted in the empirical application. In the new version Table 5 present results for labour productivity (defined as sales per worker) and table 6 for TFP. The results for value-added per worker –as an alternative measure of labor productivity– are presented in the appendix in Table A.6

4. Section 4 (model specification and main results)

a. The selection of control variables should be motivated better. For example, I don’t quite understand why the regression controls for the experience of the manager. The authors argue that their results reflect that firms with male owners only select female candidates if the latter are more qualified than comparable male candidates. Arguably, the manager’s experience is one (proxy) measure of qualification. I would therefore expect that, if the author’s hypothesis is correct, the results would be even stronger if the regressions did not control for the manager’s experience (and in fact, this would be a useful addition to the paper – see also point (d) below).

Answer: Many thanks for pointing this out, in the revised paper the models are first estimated without the variable “experience of the manager” the results are shown in Tables 5 and 6 (columns 1-3) and experience of the manager is added in column 4. The results indicate that the exclusion of the variable barely chains the results for labor productivity, whereas for TFP the core results is stronger if the regressions do not control for the manager’s experience. This evidence is discussed in the paper.

b. The numeric example quoted to illustrate the interaction effect does not match the Table. E.g., the text says “the average labor productivity is around 15 percent lower (0.219 – 0.37 = 0.152). However, as per Table 5, this should be “0.223 – 0.381 = 0.158 (=16%).

Answer: The results shown in the paper have slightly changed, because Table 5 only show results using sales per worker as dependent variable. In any case, the quoted illustration refers now to the new results and it matches the table, thanks.

c. The regional focus on MENA at the end of section 4.1 appears ad hoc and not well motivated. I understand that MENA has the lowest share of female entrepreneurs by region (as per Table 1), but the main mechanism of interest documented in the paper (i.e., the positive significant interaction between female top management and female ownership) is not significant for MENA countries. Moreover, at the country level, there are no firms with both a female top manager and female presence in ownership in Jordan and Yemen, which makes it impossible to estimate the interaction effect. I would be concerned that in some of the other countries, too, the interaction effect may be based on a small number of observations. For these reasons, the MENA deep dive provides little value added to the paper’s main storyline. Instead of zooming in on one region, it would be more useful to conduct additional analysis that could provide empirical support to the notion that women managers in firms with exclusive male ownership are disproportionately competent/qualified but not necessarily in firms with a female co-owner. One option would be to compare years of experience of male and female managers in firms with exclusive male ownership compared to firms with a female co-owner, overall and by region, and explore whether these descriptive patterns are consistent with the paper’s main hypothesis.

Answer: I agree with the argument that the specificities of the MENA region do not provided value added to the main argument of the paper. Consequently, I have revised the full paper to focus on world regions and drop the parts of the paper and the tables that referred to specific MENA countries. 

I have also considered your suggestion of comparing the average years of experience of the manager. The results indicate that women working as top managers for exclusively male owned firms have on average 13 years of experience, whereas females working for firms with female co-owners have around 16 years of experience, indicating that females are more exigent when they are among the owners. This also happened for male top managers; his average experience is higher when there are female co-owners. However, I have not included this in the paper because more experience does not necessarily mean better preparedness or better skills or education.

d. I think the paper would be considerably strengthened by attempting to connect the main results to other literature strands that are related to the key hypothesis of the paper. For example, Benson et al (2021) document that in the US retail sector women are 14 percent less likely to be promoted relative to the sample average, despite receiving higher performance ratings on average (e.g., to support the author’s argument that women, in many settings, have to be disproportionately qualified to be selected into management positions). Kunze and Miller (2017) document that women are more likely to be promoted when there are more female bosses in the next higher rank (which could be connected to the author’s argument that women owners act as mentors for female managers). I am not advocating for these particular studies to be discussed in the paper, but I do think the paper would be strengthened by providing some evidence from other contexts for the hypothesized mechanisms.

Answer: Many thanks for pointing out this interesting strand of the literature, both papers are taken on board to support the paper’s argument. They are cited in the introduction and in the literature review.

Minor comments:

5. Introduction: “Existing research on the performance gap between female and male firm’ owners for other regions” -> what is meant by “other regions”?

Answer: this refers to world regions, thank. It has been clarified

6. Section 3: “It could be argued that for female managers access to finance or crime could be a higher constraint than for male managers” -> I believe this is a hypothesis, not a result/description of the Table (because as pointed out subsequently, only Sub-Saharan Africa shows a significant gender gap in access to finance). This distinction should be made clearer. Also, rather than saying “it could be argued” it would be preferable to cite other studies that have made this argument.

Answer: Done, thanks. We refer to studies that made this argument.

7. Section 4: The text says that sales are expressed in Egyptian pounds. Is this correct? Because this would be a rather unusual reference currency for a global dataset.

Answer: All are in USD, it has been corrected, thanks.

8. Regions and acronyms: The term “South Saharan Africa” is not commonly used and should be revised to “Sub-Saharan Africa”. Likewise, the paper should use consistent acronyms and labels for world regions. For example, Latin America and the Caribbean is sometimes abbreviated LA (in section 2) and some LAC (subsequently). Sub-Saharan Africa is sometimes abbreviated SSA (mostly in the text) and sometimes AFR (in the tables). Acronyms should be spelled out on first use (which is currently not the case).

Answer: The acronyms have been homogenized, thanks.

9. The paper contains a lot of typos and would benefit from rigorous copy-editing. Some examples below (mainly from the paper’s first pages – but this continues throughout the paper)

a. “Constraint” and “constrain” are not equivalent (the first is a noun, the second one a verb). Throughout the paper, the two terms are conflated (e.g., section 2: “females face more constrains than males” -> this should be “constraints”; section 4: “factors that may constraint the performance” -> this should be “constrain”).

b. Introduction:

• “and that should be finance by the WEFI” -> financed (not finance)

• “our main contribution to the literature is the use newly available” -> of newly available

• “the methodology used consist on” -> consists

• “section 2 revise the closely related literature” -> reviews

c. Section 2

• “affect more females entrepreneurs that male entrepreneurs” –> more female entrepreneurs than male entrepreneurs

• “females socialize differently and this influence their” -> influences

Answer: A copy-editing by a native speaker has been done, thanks.

---

## [Decision Letter · Decision Letter 1]

1 Aug 2022

PONE-D-22-04064R1FEMALE TOP MANAGERS AND FIRM PERFORMANCEPLOS ONE

Dear Dr. Martínez-Zarzoso,

Thank you for submitting your manuscript to PLOS ONE. I have now heard from the referee who is happy about your revision, but suggests a number of changes, all of which seem well taken to me. Please, also keep in mind PLOS ONE’s publication criteria. Overall, we invite you to submit a revised version of the manuscript that addresses the points raised during the review process.

We look forward to receiving your revised manuscript.

Kind regards,

Joanna Tyrowicz

Academic Editor

PLOS ONE

Journal Requirements:

Reviewers' comments:

Reviewer's Responses to Questions

**Comments to the Author**

1. If the authors have adequately addressed your comments raised in a previous round of review and you feel that this manuscript is now acceptable for publication, you may indicate that here to bypass the “Comments to the Author” section, enter your conflict of interest statement in the “Confidential to Editor” section, and submit your "Accept" recommendation.

Reviewer #1: All comments have been addressed

2. Is the manuscript technically sound, and do the data support the conclusions?

Reviewer #1: Yes

3. Has the statistical analysis been performed appropriately and rigorously? 

Reviewer #1: Yes

4. Have the authors made all data underlying the findings in their manuscript fully available?

Reviewer #1: Yes

5. Is the manuscript presented in an intelligible fashion and written in standard English?

Reviewer #1: Yes

6. Review Comments to the Author

Reviewer #1: The revised paper is substantially improved. However, I still have a few minor comments, mostly related to inconsistencies between the text and tables, which the authors should review and (if necessary) correct.

P. 9:”a number similar to the average in high-income OECD and non-OECD countries, slightly lower in SSA (28 percent).” -> According to Table 1, the SSA estimate is 29 percent.

P. 9: “Finally, the third gender variable, female dominated ownership, is shown in columns 3 and 6 for world regions.” -> I don’t see a column 6 in Table 1.

P. 10: “In terms of total sales, a significantly higher number of firms are managed by males in most regions, except SAR.” -> I don’t understand this sentence. My reading of Table 3 is that male-managed firms have higher sales than female-managed firms in most regions, except SAR.

P. 10: “In Table 4, we show the participation of women in firms classified by employment size. It can be observed that a woman’s presence in ownership is more common in small and medium firms in both developed and developing countries, whereas in the MENA region, the reverse is the case.” -> My reading of Table 4 is that “female presence in ownership is *not* more common in small and medium firms in developing countries” (i.e., small = 29.8%; medium=32%; large=36%). Also, the text should be rephrased to make it clear that MENA results are not actually shown in the Table.

P. 14: The paper argues that “if females are among the owners and the top manager is a female, the average labor productivity is only around 1.2 percent lower (0.098-0.110=-0.012) according to column (3).” -> it would be useful to test if the combined effect (i.e., 0.012) is statistically different from zero.

P. 15: “The first interesting outcome is that female presence in ownership when the top manager is a male shows a positive and significant effect on labor productivity in SSA, MENA, and SAR regions and a negative effect on ECA.” -> There is also a negative effect in EAP (marginally significant).

P. 21: How is labor productivity defined in Tables 2 and 3? Is this sales per workers? If yes, then it should be labeled as such since value added per worker is also a measure of labor productivity (and is described as such in the text, e.g. p. 10: “In terms of value-added per worker (labor productivity), no significant differences are found for the EAC, SSA, and EAP regions, whereas for LAC and MENA, the performance is higher for male-managed firms.”). Tables 2 and 3 should use the same variable names.

7. PLOS authors have the option to publish the peer review history of their article (what does this mean?). If published, this will include your full peer review and any attached files.

Reviewer #1: No

---

## [Author Response · Author response to Decision Letter 1]

15 Aug 2022

Response to Review Comments to the Author

Reviewer #1: The revised paper is substantially improved. However, I still have a few minor comments, mostly related to inconsistencies between the text and tables, which the authors should review and (if necessary) correct.

P. 9: ”a number similar to the average in high-income OECD and non-OECD countries, slightly lower in SSA (28 percent).” -> According to Table 1, the SSA estimate is 29 percent. - Thanks, corrected.

P. 9: “Finally, the third gender variable, female dominated ownership, is shown in columns 3 and 6 for world regions.” -> I don’t see a column 6 in Table 1. 

- Thanks, I have deleted “and 6”.

P. 10: “In terms of total sales, a significantly higher number of firms are managed by males in most regions, except SAR.” -> I don’t understand this sentence. My reading of Table 3 is that male-managed firms have higher sales than female-managed firms in most regions, except SAR. 

- The sentence has been modified accordingly.

P. 10: “In Table 4, we show the participation of women in firms classified by employment size. It can be observed that a woman’s presence in ownership is more common in small and medium firms in both developed and developing countries, whereas in the MENA region, the reverse is the case.” -> My reading of Table 4 is that “female presence in ownership is *not* more common in small and medium firms in developing countries” (i.e., small = 29.8%; medium=32%; large=36%). Also, the text should be rephrased to make it clear that MENA results are not actually shown in the Table. 

- Many thanks for the suggested correction, it has been taken on board and the MENA part has been deleted.

P. 14: The paper argues that “if females are among the owners and the top manager is a female, the average labor productivity is only around 1.2 percent lower (0.098-0.110=-0.012) according to column (3).” -> it would be useful to test if the combined effect (i.e., 0.012) is statistically different from zero.

- Thanks for this suggestion: A different regression has been estimated to directly obtain the coefficient of the combined effect and the corresponding standard error, and the coefficient is significant at the 5% level. A footnote (7) has been added in the revised manuscript.

P. 15: “The first interesting outcome is that female presence in ownership when the top manager is a male shows a positive and significant effect on labor productivity in SSA, MENA, and SAR regions and a negative effect on ECA.” -> There is also a negative effect in EAP (marginally significant). 

- I have added this fact also in the text, thanks.

P. 21: How is labor productivity defined in Tables 2 and 3? Is this sales per workers? If yes, then it should be labeled as such since value added per worker is also a measure of labor productivity (and is described as such in the text, e.g. p. 10: “In terms of value-added per worker (labor productivity), no significant differences are found for the EAC, SSA, and EAP regions, whereas for LAC and MENA, the performance is higher for male-managed firms.”). Tables 2 and 3 should use the same variable names.

- Both changes have been made (labor productivity has been replaced by sales per worker and the variables in Table 3 are now taken from Table 2, thanks.

---

## [Editor Report · Decision Letter 2]

19 Aug 2022

FEMALE TOP MANAGERS AND FIRM PERFORMANCE

PONE-D-22-04064R2

Dear Dr. Martínez-Zarzoso,

We’re pleased to inform you that your manuscript has been judged scientifically suitable for publication and will be formally accepted for publication once it meets all outstanding technical requirements.

Kind regards,

Joanna Tyrowicz

Academic Editor

PLOS ONE
---

## [Editor Report · Acceptance letter]

24 Aug 2022

PONE-D-22-04064R2 

FEMALE TOP MANAGERS AND FIRM PERFORMANCE 

Dear Dr. Martínez-Zarzoso:

I'm pleased to inform you that your manuscript has been deemed suitable for publication in PLOS ONE. Congratulations! Your manuscript is now with our production department. 

Kind regards, 

on behalf of

Professor Joanna Tyrowicz 

Academic Editor

PLOS ONE